# Between Social Protests and a Global Pandemic: Working Transitions under the Economic Effects of COVID-19

**Valentina Rivera [1]** and **Francisca Castro [2,*]**

1. Bielefeld Graduate School in History and Sociology, Bielefeld University, D-33501 Bielefeld, Germany; valentina.rivera@uni-bielefeld.de
2. Department of Social Sciences, Humboldt University, 10117 Berlin, Germany
* Correspondence: francisca.castro@hu-berlin.de

**Abstract:** Emerging research on the economic consequences of the COVID-19 pandemic draws attention to the labor effects of the crisis in the Global South. Developing countries show high levels of labor informality, where most workers cannot work from home and depend on daily income. In addition, the scarce and late state aid makes it difficult for workers to cope with the economic hardships caused by the pandemic. This research explores the employment trajectories of workers throughout the ongoing pandemic in Chile: a neoliberal country with a strong male breadwinner culture and high levels of income inequality. Using longitudinal non-probabilistic data for Chilean employment, this study finds that men lost their jobs to a lesser extent and returned to the labor market faster than women. Likewise, male workers with family (with a partner and young children) remained employed in a higher proportion than female workers with family, and most of these women shifted from employment into care work. The existing literature already pointed out how economic crises can have adverse effects on progress towards gender equality, and the current economic crisis seems to be no exception. Labor informality and low-skilled jobs were highly related to unemployment during the first months of COVID in Chile. These are important variables in a developing economy such as Chile, where around one-third of the population works under these conditions. This article concludes by reflecting on the importance of addressing the present crisis and future economic recovery with a gender perspective.

**Keywords:** female employment; labor market; COVID-19; economic recession; unemployment during crisis

## 1. Introduction

The COVID-19 pandemic has caused not only a worldwide health emergency, but also an unprecedented global economic crisis driven by the established measures to stop the spread of the virus, such as social distancing, quarantine, and lockdowns (International Monetary Fund 2020). The economic crisis has particularly hit countries with fragile labor protection systems, such as the U.S., Brazil, and the U.K., deepening existing inequalities of race, class, and gender (Blundell et al. 2020; Nunes 2020; Ortega and Orsini 2020). State aid has been one of the main policies to help unemployed workers buffer the economic consequences of the pandemic, even in typically non-interventionist states, such as the U.S. (Adams-Prassl et al. 2020). However, in developing regions such as Latin America, state aid for workers affected by the pandemic has been mostly too little and too late (Benítez et al. 2020; Taylor 2020). Emerging research on the economic consequences of the COVID-19 pandemic expects that the economic crisis will not be the same all over the world, and that the recovery will be slower in developing countries. Countries in the Global South often count on inadequate health infrastructure and display high labor informality since most workers cannot work from home and depend on daily income (Dingel and Neiman 2020; Saltiel 2020). Thus, it is important to investigate how workers in developing countries are navigating the consequences of the pandemic.

The current crisis is especially detrimental for women. During 2020, intimate partner violence against women increased globally because of lockdowns and stay-at-home policies (Agüero 2021; Moreira and Costa 2020; Roesch et al. 2020). These policies also increased gender differences in the division of paid and unpaid work, as reported by recent studies (Adams-Prassl et al. 2020; Alon et al. 2020; Reichelt et al. 2020). Previous research based on real-time surveys has found that, during 2020, women were more likely to increase their load of housework (Del Boca et al. 2020), lose their jobs (Farré et al. 2020), and reduce their working hours more than men did (Collins et al. 2020), especially women with small children (Hipp and Bünning 2020). Evidence for working women from non-Western countries is still scarce; therefore, it is worth to study not only how workers in developing countries are coping with the present economic crisis, but also how women in the Global South are navigating the effects of the COVID-19 pandemic and its economic backlash.

Of special interest is the case of Chile, a Latin American country struck by a movement of social unrest since October 2019 (Somma et al. 2020) and, soon after, harshly hit by the COVID-19 pandemic. Throughout the last 30 years, Chile has fed an international reputation of strong macroeconomic success and stability, while generating extreme internal inequalities between an elite class and low-income households (Han 2012), as well as large gender inequalities in employment and earnings (Contreras and Plaza 2010; Ferrada and Zarzosa 2010; Perticará and Bueno 2009). As of 2018, Chile was the second most unequal country in the OECD after Costa Rica, with a Gini index of 0.46, where the richest 20% of the population earned 10.3 times more than the poorest 20%[1]. Chile has always had a historically low female labor participation rate, despite having one of the highest female education rates in the region (CEPAL 2021; Contreras and Plaza 2010; Instituto Nacional de Estadísticas 2015). Within the labor market, gender inequalities are further reproduced, as the higher the educational level, the greater the gender wage gap[2]. These rooted and pervasive inequalities ignited the massive protests in 2019, and the pandemic is further deepening the distance between the precarious and well-off workers, and between working women and men. During 2020, the female labor force declined by 41% in Chile, and studies calculate that the first year of the pandemic caused ten years of setbacks for working women[3].

This article aims to study the changes in employment during the COVID-19 pandemic in Chile, particularly in female employment. For this purpose, we drew on panel data from the National Employment Survey (*Encuesta Nacional de Empleo*, ENE) conducted by the country's Institute of Statistics (*Instituto Nacional de Estadísticas*, INE). This article is mainly exploratory, since it describes the different labor trajectories of men and women during the pandemic. Using sequence analysis, we explore the trajectories on a sample of workers from January to November 2020, examining the continuities, breaks, and halts in their employment. We found that men remained employed at a higher rate during the pandemic and recovered their jobs faster than women did. Likewise, men who became unemployed moved into joblessness, while women went mostly into inactivity for family reasons. Married men, men with children, and men who are their household's main breadwinner remained employed at higher rates, while women who transited into long inactivity were less educated, with worse working conditions, in a relationship (married or cohabiting), and with children, becoming economically dependent on their partners. Previous studies have shown that economic crises have adverse effects on progress toward gender equality (Karamessini and Rubery 2013; Sani 2018), and the effects of the ongoing pandemic into the economic sector have not been the exception.

---

[1]    OECD Income Distribution Database (IDD) Available online: https://stats.oecd.org/Index.aspx?DataSetCode=IDD (accessed on 9 April 2021).

[2]    Supplementary Income Survey applied by the Chilean National Institute of Statistics. Available online (only in Spanish): https://www.ine.cl/docs/default-source/encuesta-suplementaria-de-ingresos/publicaciones-y-anuarios/s%C3%ADntesis-de-resultados/2018/sintesis_nacional_esi_2018.pdf?sfvrsn=eed2fa51_3 (accessed on 9 April 2021).

[3]    Desigualdad de género: un año de pandemia, diez años de retroceso para las mujeres. *Diario Universidad de Chile*. Available online (only in Spanish): https://radio.uchile.cl/2021/03/08/desigualdad-de-genero-un-ano-de-pandemia-diez-anos-de-retroceso-para-las-mujeres/ (accessed on 3 March 2021).

This research contributes to the current literature on female labor market participation in two ways. First, it offers a new perspective on the effects of the pandemic in the Global South, as most of the studies focus on Western economies. A better understanding of how the labor market responds to economic crisis in less developed, gig-oriented economies could help not only academics but also policymakers in understanding potential outcomes of the recession. Second, through the analysis of longitudinal panel data, we can more accurately assess the effects of the pandemic in our interest group (women who are part of the labor market before the pandemic). Longitudinal studies can examine how precarious work creates vulnerabilities in different domains, helping to prevent outcomes of the COVID-19 pandemic, such as unemployment (Blustein et al. 2020). Despite the ENE Survey being more appreciated by policymakers and technocrats and, on the contrary, overlooked by scholars, this data source is extremely valuable. As stated by Blustein et al. (2020), statistical analyses such as cluster analysis from distinct groups of unemployed individuals can help both researchers and policymakers to identify the most suitable intervention for each group. We believe this research could represent a first step in considering this data source as a valuable asset for information.

## 2. Theory and Context

### 2.1. Female Employment in Times of Economic Crises

Gender differences in vulnerability to recession and austerity derive from differences in women's position in the job structure, relative to men's (Karamessini and Rubery 2013; Çağatay and Özler 1995). For instance, men predominate in management, blue-collar crafts, transportation, and construction; meanwhile, women dominate professions in the service sector (England 2005). Their concentration in counter-cyclical sectors protects women from unemployment, as they are not affected directly by common recessions. On the other side, male employment is on the front line of job destruction (Périvier 2014), which is one of the reasons why economic recessions were often called "man-cessions" (Bredemeier et al. 2017; Perri and Steinberg 2012; Strolovitch 2013), as men are more likely to lose their jobs during economic crises than women, who most of the time become the "shock absorbers" of crises (Beneria and Feldman 1992; Chant 1994; Hite and Viterna 2005). For example, during the Indonesian crisis, many women joined family business, making a major contribution to mitigating the impact of such a crisis on family income (Thomas et al. 2000). During the East Asian crisis, male unemployment worsened more than female employment because of the bigger decline in the industrial and trade sector; meanwhile, the female-intensive service sector was not so badly affected (Lim 2000). During the same crisis, countries such as Korea responded with an increase in female participation in the labor market. However, downward occupational moves were more common during this period (Ma 2014), suggesting that, even when female labor participation can increase during times of crisis, it is usually in worse conditions than non-crisis times.

The recent economic crisis of 2008–2009 affected mainly cyclical occupations where most men work (Bredemeier et al. 2017), but in the short term, male employment recovered more quickly than female employment (Taylor et al. 2011). Researchers on the gender impact of economic crisis have stated that the financial crisis of 2008 had adverse effects on the progress towards inequality in Europe (Alcañiz and Monteiro 2016; Hozic and True 2016; Sani 2018), especially because the austerity programs proposed by European and U.S. governments to face the crisis ended up undermining important employment and social welfare protections that ensured gender equality (Hermann 2017; Karamessini and Rubery 2013).

Research on the effects of the COVID-19 pandemic on employment claims that this recession will be no "man-cession", as the measures to stop the spread of the virus, especially quarantines and lockdowns, affected the service sector in higher proportions, where most women are employed (Adams-Prassl et al. 2020; Alon et al. 2020; Blundell et al. 2020; Dang and Nguyen 2021). This is especially relevant for the case of Chile, considering the structure of occupational groups: 26.6% of employed women in Chile work in "non-qualified jobs", and 23.4% of them work as service and sales workers (Instituto

Nacional de Estadísticas 2015). Early studies on the impact of the current pandemic on employment in the U.S. found that workers with less education, service workers, women, part-time workers, and racial and ethnic minorities concentrated the largest unemployment rates (Couch et al. 2020; Falk et al. 2021). Evidence from other countries shows that losing employment would be particularly detrimental for women in lower occupations (Cortes and Forsythe 2020; Kikuchi et al. 2021). Thus, we hypothesize that:

**Hypothesis 1.** *In Chile, women in low-skilled occupations and/or with poor working conditions will be the most affected by unemployment during the pandemic.*

The gender impact of economic crises is also linked to the distinctive positions that men and women occupy in the family economy and in the welfare system. The allocation of these positions is largely related to the norms and values existing around gender relations and the division of paid and unpaid work (Korpi 2000; Lewis 1992; Pfau-Effinger 2005), where women are often assumed to be carers first and labor force participants second, acting as voluntary or involuntary labor reserve (Bruegel 1979). The "flexible" nature of female employment would encourage them to work during economic upturns, but, in turn, they are expelled from the labor market during recessions (Périvier 2014). Additionally, because of the care penalty, women increase their presence in atypical and precarious jobs, occupying part-time jobs and temporary contracts with a higher incidence (Pérez Ortiz et al. 2020).

The COVID-19 pandemic not only harshly affected female employment, but it also reinforced their roles as caregivers due to both the closure of daycare centers and schools around the world and the "stay-at-home" measures that have incremented their loads of unpaid work (Del Boca et al. 2020; Farré et al. 2020; Hipp and Bünning 2020; Mussida and Patimo 2020). Due to this, as early as March 2020, the Coronavirus pandemic was called "a disaster for feminism"[4]. Thus, we also hypothesize that:

**Hypothesis 2a.** *Partnered women with young children will be the most affected by unemployment during the pandemic, moving from paid work to unpaid care work.*

**Hypothesis 2b.** *Conversely, partnered men with young children will be the most likely to maintain their employment, given the strong male breadwinner culture present in Chilean society.*

Below, we review the particularities of female employment in Chile and how the ongoing Coronavirus pandemic is affecting the relationship between women and work.

*2.2. Female Employment in Chile*

During 1990–2010, Chile was the country with the lowest labor force participation rate in South America, as seen in Figure 1, despite the country's steady economic growth and declining fertility rate during the last century. Women in Chile are more educated and perform better than men (Bravo et al. 2008), but their labor participation is significantly lower than their male counterparts (Mora 2013). Some studies have called this contradiction "the Chilean exception" or the "Chilean gender paradox" (Ferrada and Zarzosa 2010; Fort et al. 2007; Larrañaga 2006), as the relatively traditional values and attitudes regarding gender roles, particularly the mother's role as the principal caregiver, might be the major causes of this employment setback (Contreras and Plaza 2010; Murray 2015).

---

[4]    The Coronavirus Is a Disaster for Feminism. *The Atlantic*. Available online: https://www.theatlantic.com/international/archive/2020/03/feminism-womens-rights-coronavirus-covid19/608302/ (accessed on 2 April 2021).

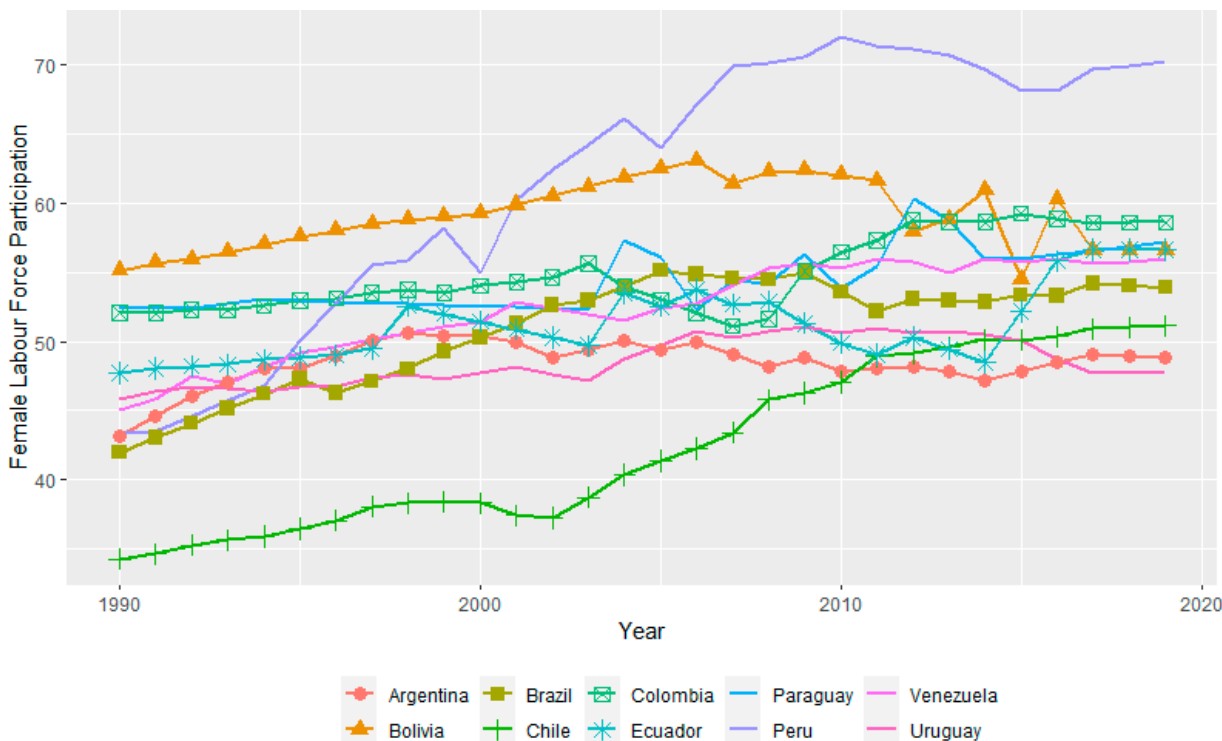

**Figure 1.** Female labor force participation rate 1990–2019, by South American countries. Source: Own elaboration based on the ILOSTAT database from the International Labor Organization.

In the last few years, consecutive governments have tried to address this issue, implementing several reforms and public policies aiming to improve the economic and social position of women in Chile. These reforms included the Divorce Law (2004), the implementation of an early childhood protection system that massively expanded childcare services (Chile Crece Contigo, 2006), and the extension of the maternity paid leave from 3 to 6 months, as well as the possibility of sharing six weeks with the father. However, these policies have not been paired with changes in the Chilean traditional gendered work culture that does not tolerate non-standard working arrangements such as part-time employment, discouraging the long-term attachment of female careers (Cabello-Hutt 2020; Madero-Cabib et al. 2019b).

During the 21st century, Chile has not experienced major economic crises, maintaining a steady economic growth and reducing its poverty rate. This economic growth has not been accompanied by a decrease in income inequality: the country still has a relatively low level of redistribution, and fiscal policy has a limited capacity for reducing extremely high market inequalities (OECD 2015), making Chile one of the most unequal countries in both Latin America and among developed countries (Flores et al. 2020). This persisting inequality is the main reason for the emergence of social protests that started in October 2019, under the slogan "*No son 30 pesos, son 30 años*" (It's not 30 pesos, it's 30 years)[5]. The economic crisis derived from the COVID-19 pandemic may the biggest economic crisis that Chile has faced since the Asian crisis in 1999, widening the gap between skilled and unskilled workers, and between women and men.

According to the National Institute of Statistics, unemployment in Chile peaked in May–July 2020 with a 13.1 percent national unemployment rate, as seen in Figure 2. The decrease in employment mainly affected the trade sector (−24.9%), construction

---

[5]　This is related to the event that acted as a catalyst for the protests that started in October: the subway fare increase of 30 pesos in the Metropolitan Region. The protests continued despite the revocation of this unpopular measure, under this slogan that points to the fact that it was not this particular event that ignited social unrest, but the persistent inequality of the last 30 years.

(−34.8%), and accommodation and food service (−49.5%)[6]. Likewise, the number of workers decreased by 11.7 percent throughout 2020[7]. Unemployment peaked higher for men than for women during the first months of the pandemic (April to September 2020), but it swiftly decreased once the lockdown measures were relaxed, as Figure 2 shows. In contrast, female unemployment in Chile appears to be more resistant to decline. It is worth mentioning that the unemployment rate only considers those people looking for work, dismissing the population that becomes inactive for family or care reasons (who are mainly women). This study aims to study labor trajectories during the pandemic in Chile, distinguishing between the different states of unemployment (e.g., inactivity because of joblessness or inactivity because of family reasons), which allows us to capture the variation between women's and men's employment trajectories and the long overlooked care work of women. This leads us to our third hypothesis:

**Hypothesis 3.** *Women who became unemployed during the pandemic will be inactive for family and care-giving reasons at a higher rate than men.*

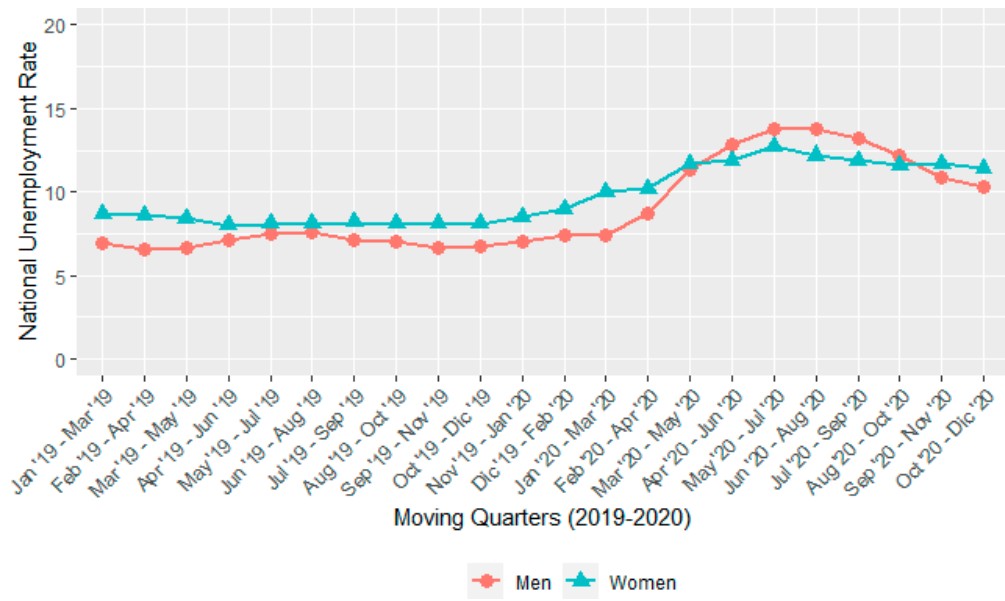

**Figure 2.** National Unemployment Rate 2019–2020, by women and men. Source: Own elaboration based on ENE.

During the pandemic, state transfers to help unemployed workers have been scarce, of low quantity, and highly targeted[8]. The only significant universal policy to relieve the economic consequences of unemployment was the approval of a constitutional reform that allowed an exceptional withdrawal of 10 percent of the accumulated pension funds from individual capitalization in late July, which was extended with a second withdrawal of another 10 percent in December 2020. This twenty percent pension withdrawal is expected to jeopardize the future economic security and retirement of all workers in Chile[9].

6　Boletín Estadístico Empleo Trimestral Diciembre 202. Available online: https://www.ine.cl/docs/default-source/ocupacion-y-desocupacion/boletines/2020/pa%C3%ADs/bolet%C3%ADn-empleo-nacional-trimestre-m%C3%B3vil-mayo-junio-julio-2020.pdf?sfvrsn=b6bcfa71_4 (accessed on 9 April 2021).

7　Boletín Estadístico Empleo Trimestral Agosto 202. Available online: https://www.ine.cl/docs/default-source/ocupacion-y-desocupacion/boletines/2020/pa%C3%ADs/bolet%C3%ADn-empleo-nacional-trimestre-m%C3%B3vil-mayo-junio-julio-202.pdf?sfvrsn=b6bcfa71_4 (accessed on 9 April 2021).

8　Por qué el plan económico ante el Covid-19 es insuficiente y lo agrava el dictamen que permite no pagar remuneraciones. *Ciper Chile*. Available online: https://www.ciperchile.cl/2020/03/27/por-que-el-plan-economico-ante-el-covid-19-es-insuficiente-y-lo-agrava-el-dictamen-que-permite-no-pagar-remuneraciones/ (accessed on 11 February 2021).

9　Sin dogmas ni trincheras: ¿cuál es el real impacto del retiro del 10% de las AFP? *El Mostrador*. Available online: https://www.elmostrador.cl/destacado/2020/07/20/sin-dogmas-ni-trincheras-cual-es-el-real-impacto-del-retiro-del-10-de-las-afp/ (accessed on 11 February 2021).

Chilean women are especially disadvantaged in the Chilean mandatory IRA (Individual Retirement Account) scheme, as they have more unstable work trajectories than men or did not take part at all in the labor market during their life course (Cabello-Hutt 2020; Madero-Cabib et al. 2019a) because of the strong male breadwinner culture prevailing in the country (Contreras and Plaza 2010; Staab 2012), consequently having little or no savings in their pension's account (Madero-Cabib et al. 2019b). Some argue that once the cause of the recession has passed, the economy may quickly return to normal (Weinstock 2020); however, the longer the pandemic and resulting recession last, the more likely certain effects are to be longer lasting as well, as shown by previous research on the effects of economic recessions in the European Union, where the crisis had more perverse effects in those countries where the duration and intensity of the crisis were higher and where strong structural adjustment measures were implemented (Pérez Ortiz et al. 2020). Therefore, it is yet to be seen what shall be the consequences in both the short and long term of the pandemic on women's employment in Chile.

## 3. Design

### 3.1. Data

This article aims to study the change in employment during the COVID-19 pandemic in Chile. Our study combines exploratory (sequence analysis) and predictive (multinomial logistic regression) methods to understand the relationship between the pandemic and the changes in employment status in Chile throughout 2020. We analyzed data from the ENE, a national-based survey conducted by the National Institute of Statistics of Chile (INE) since 2010. The ENE's main purpose is to measure the unemployment rate in the country, providing estimates of market indexes for mobile quarters, such as labor informality, working hours, and employment relationships, among other indicators appropriate for the present study.

The ENE is a panel survey with a rotating sample design, where the units of analysis are interviewed more than once at several points in time. Due to the COVID-19 pandemic, face-to-face surveys were suspended in March 2020 and replaced by telephone data collection, which increased the loss of sample sizes. The INE has conducted several methodological exercises to calculate potential estimation biases, without showing statistically significant biases in the estimation of employment and unemployment rates[10]. We emphasize the continuous efforts of the INE to ensure the continuity of data collection, guaranteeing the statistical quality of its estimates by following the international recommendations of the ECLAC and the ILO (Instituto Nacional de Estadísticas 2020).

To study the change in employment during 2020, we relied on a sub-sample of individuals of working age[11] who declared to be working before the first case of COVID-19 in Chile[12], and were subsequently surveyed after the implementation of sanitary measures such as quarantines, curfews, store closures, and movement restrictions in public places. The main criteria for case selection were: (1) working people surveyed before the start of the pandemic in March (that is, surveyed during January or February 2020), and (2) people successfully interviewed at least three additional times during 2020, for a total of four waves. This resulted in an N of 8864 cases (35,456 people/month), as depicted in Table 1.

---

[10] Nota Técnica N°5, Instituto Nacional de Estadísticas. Available online (only in Spanish): https://www.ine.cl/docs/default-source/ocupacion-y-desocupacion/publicaciones-y-anuarios/separatas/tem%C3%A1ticas/separata-t%C3%A9cnica-n-5-contingencia-covid-19.pdf (accessed on 9 April 2021).

[11] 18–59 years for women, and 18–64 years for men, as the retirement age in Chile is of 60 years old for women and 65 years old for men.

[12] The first case of COVID-19 in Chile was confirmed on March 3, 2020. See note available online: https://www.reuters.com/article/us-health-coronavirus-chile/chile-records-first-confirmed-case-of-coronavirus-health-ministry-idUSKBN20Q2UU (accessed on 16 April 2021).

**Table 1.** Summary of ENE waves considered for analysis.

|  | Months | Observations |
|---|---|---|
| Wave 1 | January–February | 8864 |
| Wave 2 | April–May | 8864 |
| Wave 3 | July–August | 8864 |
| Wave 4 | October–November | 8864 |
| | N = 8864 | N-months = 35456 |

As the aim of this study is not causal but descriptive, we will not resort to the use of weights, focusing on the internal validity of the study and trying to minimize the loss of sample cases.

*3.2. Analysis Plan*

3.2.1. Sequence Analysis: Determining Working Trajectories

In the first section, we performed sequence analysis on the sub-sample of men and women who declared to be working after the pandemic. We constructed the employment statuses to be sequenced as trajectories based on the variable "economic activity status" available in the survey. These employment statuses are: (1) *working*, (2) *on furlough*[13], (3) *unemployed* (either actively looking for a job or inactive reasons other than care work), and (4) *care work* (inactive for personal or familiar reasons). We used the R packages TraMineR (Gabadinho et al. 2011) and WeightedCluster (Studer 2013) to perform the analysis.

Sequence analysis chronologically allocates individual trajectories of employment, allowing the researcher to explore intra-individual variability across time. Then, by clustering these sequences, we can identify certain types of trajectories that are homogeneous and distinct from one another.

To cluster these trajectories, we computed the pair wise optimal matching (OM) distances between the sequences. Optimal matching calculates the minimal cost needed to transform one sequence into another through two operations: the substitution of an element of a sequence with a different element, or the insert/deletion ("indel") of an element from the sequence (Gabadinho et al. 2011; MacIndoe and Abbott 2009). As the cost of transiting to a different status is not the same (i.e., it is not the same to transit from working to on furlough than from jobless to inactive), we constructed a custom cost matrix, where we included a higher cost to transit from on furlough to the other categories and vice versa, as "On furlough" is a special category of those employed, and it is more costly for employers to fire someone protected by the furlough. We tested different cost matrices, such as robustness exercises (not present in this article but available upon request), and found that the clusters with the best fit measures and explanatory potential for the questions proposed by the present work derived from the matrix employed.

With the obtained OM distance matrix, we performed a hierarchical cluster analysis (Ward) to identify typologies of sequences. We performed the cluster analysis separately for women and men, as the literature states that they follow different employment trajectories (Ponomarenko 2016).

Figures A1 and A2 in Appendix A show the standardized quality measures for up to 10 clusters for men and women. To determine the most appropriate number of sequences, we considered both the measures of the quality of a partition proposed by Studer (2013): average silhouette width weighted—(ASWw), Hubert's γ (HG), point biserial correlation

---

[13] This employment status was introduced by the Law of Employment Protection, which allows employers to either suspend the contract of the employees by "authority act" (that is, when the authority obliges the partial or complete cessation of activities due to, for example, quarantines), or to pact an agreement with the employer for the suspension of the contract, which allows the employee to have access to their unemployment insurance. This law was mostly created to protect employers, not employees, leaving the latter in limbo between employment and unemployment. For more information, visit https://www.afc.cl/ley-proteccion-al-empleo/suspension-del-contrato-laboral (accessed on 9 April 2021).

(PBC), and Hubert's C (HC)[14], as well as those categories most useful for the analysis. Therefore, we stood with 4 and 3 clusters for women and men, respectively.

3.2.2. Multinomial Logistic Regression: Exploring the Likelihood of Cluster Membership

After clustering the workers in their respective working trajectories, we moved to the second analysis section. Here, we estimated a multinomial logistic regression to predict the probabilities of falling into each category based on demographic characteristics for women and men collected during the first wave. We provide details of these variables in the section below.

*3.3. Variables*

To study how workers in each trajectory differed from one another, we considered the following covariates:

- Sociodemographic variables such as *age*, *gender*, *educational level* (a binary variable with 1 = if the respondent has higher education, and 0 = otherwise), and *finished studies* (a binary variable with 1 = if the respondent completed their highest level of education, and 0 = otherwise)
- Family variables such as *relationship status* (a binary variable with 1 = if the respondent lives with a partner -either married or cohabiting-, and 0 = otherwise), *presence of children 0 to 5 years*, *presence of children 6 to 13 years*, and *breadwinner* (a binary variable with 1 = if the respondent declared to be the household's sole breadwinner and 0 = otherwise).
- Job characteristics such as *employment relationship* (a binary variable with 1 = if the respondent declared to be an self-employed worker, and 0 = otherwise), *working hours* (a binary variable with 1 = if the respondent declared to work 30 hours or less, and 0 = otherwise), *employment formality* (a binary variable with 1 = if the respondent is an informal worker—i.e., they have neither social security nor pension money in their retirement fund, and 0 = otherwise), *employment contract*: (a binary variable with 1 = if the respondent has a contract, and 0 = otherwise), and *ISCO Skill Level* (a 3-category variable derived from the Occupational Classification ISCO-88 with 1 = high-skilled occupation, 2 = medium-skilled occupation, and 3 = low-skilled occupation)
- Economic sector, a 4-category variable with 1 = primary sector, 2 = manufacture and construction, 3 = commerce, and 4 = services.
- Additional controls such as region (metropolitan, north, and south) and sector (urban, rural)

Table 2 shows a summary of the descriptive variables. The $\chi^2$ test in the third column measures the independence between the exposed variables and gender, reporting whether there is an association between the respective variables at the different confidence levels specified.

Table 2 illustrates that employed women in our sample were younger and more educated than men. They were also more likely to live without a partner and with small children than men do. However, they were less prone to declare being the main breadwinner of the household. Regarding their employment arrangements, women worked more in part-time schemes than their male counterparts, and accordingly, they worked fewer hours on average than them. They also had fewer years of seniority. There were no major gender differences in self-employment, employment contract, and labor informality, with around one-third of workers in our sample working informally (i.e., with neither social security nor pension money in their retirement fund) and/or without a contract.

---

14    **ASWw:** Coherence of assignments (between-group distances and within-group homogeneity). **HG:** capacity of the clustering to reproduce the distances (order of magnitude). **PBC:** capacity of the clustering to reproduce the distances. **HC:** gap between the partition obtained and the best partition theoretically possible. For more details see (Studer 2013).

**Table 2.** Descriptive statistics.

| | | Women (n = 3981) | Men (n = 4883) | $\chi^2$ Test |
|---|---|---|---|---|
| Age | 18–30 years old | 19% | 17% | |
| | 1–45 years old | 41% | 33% | *** |
| | 46–64 years old | 40% | 50% | |
| Education | Higher education | 42% | 32% | *** |
| | Finished studies | 80% | 74% | *** |
| Family variables | With partner | 51% | 69% | *** |
| | Children 0 to 5 | 23% | 21% | ** |
| | Children 6 to 13 | 38% | 31% | *** |
| | Main breadwinner | 42% | 70% | *** |
| Employment variables | Self-employed | 20% | 18% | |
| | Part-time | 26% | 11% | *** |
| | Formal employment | 72% | 75% | ** |
| | With contract | 65% | 67% | |
| | Mean hours working | 42.6 | 51.2 | *** |
| | Mean years seniority | 6.7 | 8.4 | *** |
| ISCO skill level | High | 34% | 23% | |
| | Medium | 42% | 57% | *** |
| | Low | 24% | 19% | |
| Economic sector | Primary sector | 9% | 23% | |
| | Manufacturing and construction | 9% | 23% | *** |
| | Commerce | 19% | 14% | |
| | Services | 63% | 41% | |

Source: own elaboration based on ENE 2020. $p < 0.001$ ***, $p < 0.05$ **, $p < 0.01$ *. "Mean hours working" and "Mean years working" are average values and not percentage.

It is important to note that women mainly work in occupations with higher skills (i.e., managers, professionals, and technicians) in higher proportions than men do. This does not mean, necessarily, that women are positioned in the upper part of the earnings distribution, or that they have access to top management positions, which, in Chile, is not the case. According to a recent study by Sánchez et al. (2020), Chilean men earn approximately 19 to 28% more than women because of the difference in labor supply elasticity. This variation in gender labor supply elasticity is explained by the differences in elasticities to/from non-employment, and because informality may be more attractive for Chilean women due to, for example, lack of childcare coverage. Likewise, over two-thirds of women work in the service sector, compared to 41% of men, a horizontal segregation that goes in the same line as previous reports in the literature (England 2005).

## 4. Results

### 4.1. Working Trajectories of Men and Women during the First Months of the Pandemic

Figure 3 shows the employment trajectories from January to October 2020 for all men and women in our sample who declared to be working before the pandemic in Chile. Each horizontal line represents the sequences followed by one person, showcasing a wide range of trajectories. Although women had a lower percentage of employment, they had more unsteady and diverse trajectories than men. We can verify this observation by looking at the entropy index. The entropy index is a measure of the diversity of states observed in each position, ranging from 0 when all cases are in the same state to 1 when all cases are in different states (Gabadinho et al. 2011). Table 3 shows the entropy index at each period in the sequence by gender. Here, we see that the entropy index of women is always higher than of men, i.e., they have more diverse states in each time position than men.

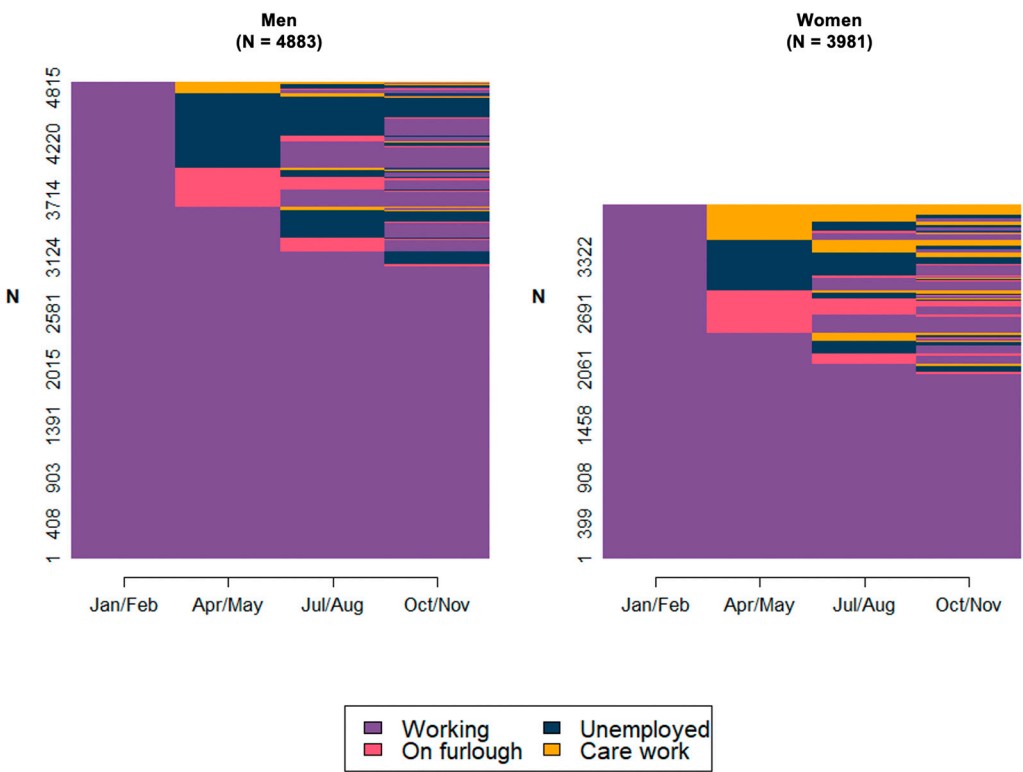

**Figure 3.** Employment trajectories during 2020 in Chile for women and men. Source: own elaboration based on ENE. The y-axis represents the frequency of cases.

**Table 3.** Entropy index at each position in the sequence, for women and men.

|  | Jan/Feb | Apr/May | Jul/Aug | Oct/Nov |
|---|---|---|---|---|
| Women | 0.00 | 0.76 | 0.73 | 0.60 |
| Men | 0.00 | 0.58 | 0.57 | 0.40 |

Source: own elaboration based on ENE.

Table 4 summarizes the employment clusters of women and men, including their respective fit measures. Three groups are present for both women and men clusters, with one additional group for women (*transition to care work*). Focusing on the fit measures, we see that the *continuously employed* cluster has a strong structure in the coherence of assignment (ASWw) for both men and women, i.e., most of them followed the same path (worked during the survey waves). The remaining clusters have a poor structure because their transitions occurred at different times, as we can see for their high entropy index. However, we prioritized clustering the trajectories according to people who shared similar transitions, instead of people who shared the same timing of the transition.

Table 4 also shows that a higher proportion of men (69%) remained continuously employed throughout the first months of the pandemic than women (55%). A higher percentage of women (18%) were on furlough when compared to men (11%), probably because women are employed to a greater extent in the service sector, one of the economic sectors most affected by the measures to halt the spread of the pandemic, as was previously mentioned. A higher percentage of men transitioned to unemployment (20%) than women did (14%). However, an important group of women stopped working and were dedicated to care work (13%). This would not be the case for men, who barely reported being inactive for personal and/or family reasons.

**Table 4.** Employment clusters for women and men.

| | Women (n = 3981) | | | | Men (n = 4883) | | | |
|---|---|---|---|---|---|---|---|---|
| | N | % | ASWw | Entropy | N | % | ASWw | Entropy |
| 1 Continuously employed | 2172 | 55% | 0.93 | 0.03 | 3353 | 69% | 0.89 | 0.05 |
| 2 On furlough | 706 | 18% | 0.19 | 0.54 | 552 | 11% | 0.18 | 0.53 |
| 3 Transition to unemployment | 576 | 14% | 0.10 | 0.55 | 978 | 20% | 0.17 | 0.48 |
| 4 Transition to care work | 527 | 13% | 0.25 | 0.56 | | | | |

Source: own elaboration based on ENE. Note: ASW (average silhouette width) measures the coherence of assignments. Interpretation: 0.00–0.25: no structure, 0.26–0.50: weak structure, 0.51–0.70: reasonable structure, 0.71–1: strong structure., see Kaufman and Rousseeuw (2009). Entropy: 0 (all cases are in the same state) – 1 (all cases are in different states).

Figure 4 plots the employment trajectories for men and women in our sample during 2020, and Table 5 summarizes the descriptive statistics for each cluster. We describe each trajectory in detail below.

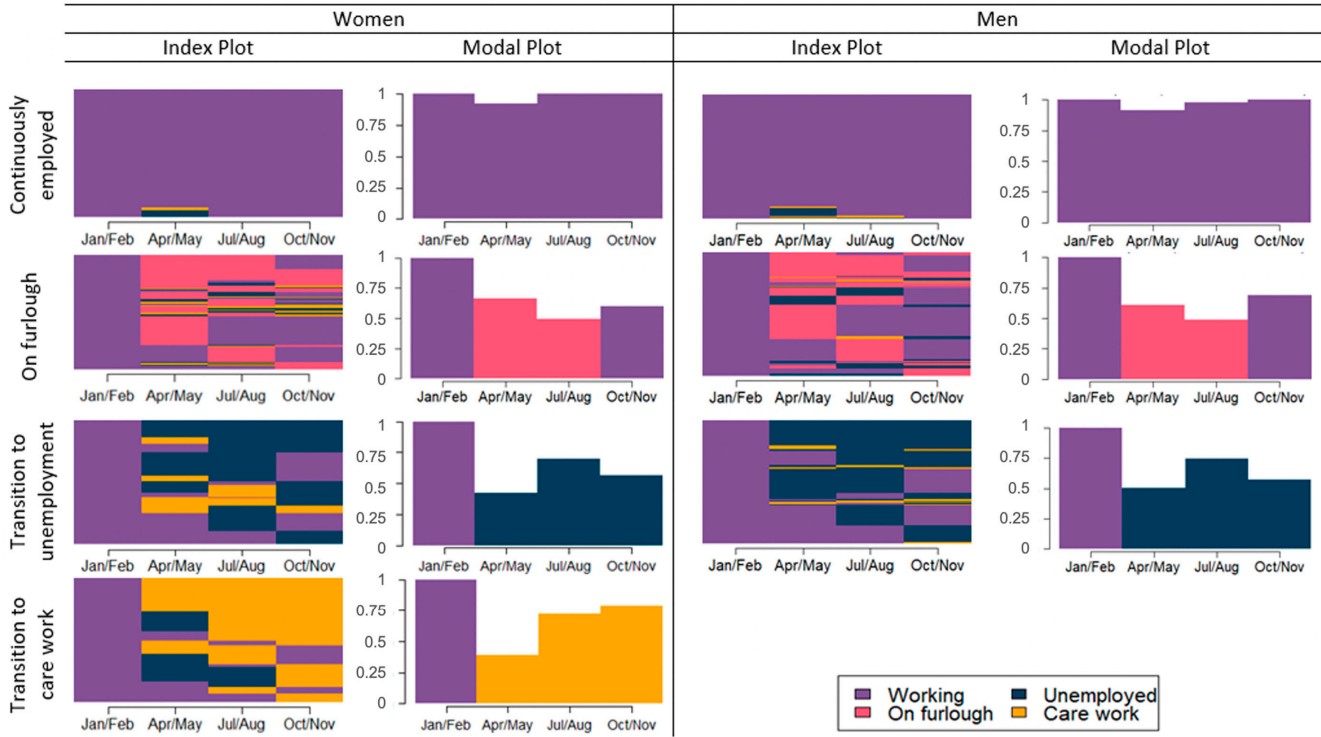

**Figure 4.** Employment cluster for men and women during 2020. Source: own elaboration based on ENE 2020. Note: the index plot represents the sequences followed by one worker, meanwhile the modal plot represents the modal status vertically for each wave. Y-axis on the modal plot represents the percentage of cases.

**Table 5.** Descriptive statistics for the clusters of women and men.

| | | Women | | | | Men | | |
|---|---|---|---|---|---|---|---|---|
| | | Continuously Employed | On Furlough | Transition to Unemployment | Transition to Care Work | Continuously Employed | On Furlough | Transition to Unemployment |
| | 18–30 | 18% | 17% | 25% | 19% | 16% | 14% | 22% |
| Age | 31–45 | 44% | 39% | 33% | 40% | 36% | 27% | 25% |
| | 46–64 | 38% | 44% | 42% | 41% | 48% | 59% | 52% |
| Education | Higher education | 53% | 37% | 25% | 17% | 36% | 26% | 22% |
| | Finished education | 85% | 80% | 73% | 67% | 76% | 76% | 64% |
| Family variables | With partner | 51% | 47% | 46% | 66% | 71% | 71% | 60% |
| | With children 0 to 5 | 23% | 22% | 19% | 31% | 23% | 18% | 17% |
| | With children 6 to 13 | 36% | 36% | 32% | 50% | 33% | 30% | 28% |
| | Main breadwinner | 46% | 48% | 33% | 27% | 74% | 73% | 57% |
| Employment variables | Self-employed | 14% | 15% | 29% | 41% | 14% | 18% | 35% |
| | Part-time | 18% | 21% | 40% | 55% | 8% | 11% | 23% |
| | Formal employment | 83% | 85% | 51% | 34% | 81% | 82% | 51% |
| | Contract | 76% | 77% | 45% | 28% | 73% | 70% | 43% |
| | Mean working hours | 41.9 | 39.9 | 34.6 | 30.3 | 45.8 | 45.3 | 40.6 |
| | Mean years of seniority | 7.7 | 7.7 | 4.2 | 3.9 | 8.9 | 9.7 | 5.9 |
| ISCO Skill Level | High-skilled occupation | 46% | 27% | 17% | 10% | 28% | 15% | 14% |
| | Medium-skilled occupation | 37% | 49% | 45% | 56% | 55% | 68% | 61% |
| | Low-skilled occupation | 18% | 24% | 37% | 34% | 18% | 17% | 25% |
| Economic Sector | Primary sector | 7% | 1% | 14% | 20% | 24% | 12% | 26% |
| | Manufact. and construction | 9% | 6% | 10% | 13% | 22% | 26% | 33% |
| | Commerce | 19% | 17% | 19% | 24% | 14% | 16% | 12% |
| | Services | 65% | 75% | 56% | 42% | 40% | 46% | 29% |
| Zone | Urban | 83% | 86% | 76% | 64% | 72% | 83% | 69% |

Source: own elaboration based on ENE 2020. "Mean hours working" and "Mean years working" are average values and not percentage.

1. *Continuously employed:* Not only did more men than women remain employed during 2020, but also men in this cluster had a partner (71%) and were the main breadwinner of the household (74%) at a higher rate than their female counterparts, as Table 5 shows. This tendency might be related to the male breadwinner culture long ingrained in Chilean society (Contreras and Plaza 2010; Murray 2015; Staab 2012). Both men and women who remained continuously employed throughout 2020 had good working conditions, i.e., low self-employment, low labor informality, and most of them worked under contracts. They were also the most educated groups, with women having a higher percentage of higher education (53%) than men (36%). As Rama (2015) has argued, in Latin America, the high return rates of professional work and the lower unemployment are traditionally experienced by those workers with higher educational attainment, as in this case. The main gender difference in this cluster is that women who remained continuously employed during 2020 worked primarily in high-skilled occupations (46%), i.e., as managers, professionals, technicians, etc., and were highly concentrated in the service sector (65%), while men in this cluster mainly worked in medium-skilled occupation (55%), and were evenly distributed throughout the economic sectors. This would indicate that women in more privileged and higher positions could continue working without interruptions during the pandemic in higher proportions than less privileged, less educated women.

2. *On Furlough*: As we stated before, women in our sample moved into furlough at a higher percentage (18%) than men did (11%). This is a less-educated group than the previous one (although women still hold a greater rate of higher education than men), but with the same working conditions and family characteristics as the first cluster. The main difference is that women and men who moved into furlough are highly concentrated in medium-skilled occupations (i.e., clerks and sale workers, as well as skilled workers), and in the service sector, which is the economic sector most affected by the social distancing measures to prevent the spread of the virus. This way, as the service sector concentrates a large portion of female workers, women may have moved into furlough in a higher proportion than men. It is worth noting, as Figure 4 shows, that more furloughed men returned to work for the fourth wave of the survey in October/November (65%) than women did (56%).

3. *Transition to unemployment*: The cluster of workers who became unemployed has the highest percentage of young people for both genders. They are also less educated than the previous two groups and have worse working conditions, i.e., high self-employment, high part-time jobs (especially women), high labor informality, and almost half of them working without contracts. Their most distinctive feature is that they are the group that congregates the most workers in low-skilled occupations. Although, women who moved into unemployment worked mostly in the service sector (56%), while men worked mostly the manufacturing and construction sector (33%). Figure 4 also shows that about one-third of the men and women who became unemployed during the first months of the pandemic regained their employment by the fourth wave of the survey.

4. *Transition to care work*: The fourth cluster of women not only became unemployed after the start of the pandemic, but also reported being inactive mainly for family reasons and to perform care work. Again, linked to the male breadwinner culture in Chile, women who stopped working for family reasons were, for the most part, partnered women with children, very few of them being the main household provider. They were also the group with the worst working conditions: 41% of them work as self-employed, 66% as informal workers, and only 28% work with a contract. These women with particularly poor working conditions and large family responsibilities are the ones considered by the literature as voluntary labor reserve (Bruegel 1979), assumed to be carers first and labor force participants second, and who are mostly expelled from the labor market during recessions (Périvier 2014). It is worth noting the association between poor working conditions, family responsibilities, and care-giving in this group of women, which could suggest the poor working conditions of working mothers in Chile.

## 4.2. Predictors of Working Trajectories

Table 6 also shows the results of the multinomial logistic regression. The coefficients are displayed in log odds, with continuously employed as the reference group for both women and men.

**Table 6.** Multinomial logistic models (log odds) of individual characteristics on employment trajectories (ref. = continuously employed).

| | Women | | | Men | |
|---|---|---|---|---|---|
| | **On Furlough** | **Transition to Unemployment** | **Transition to Care Work** | **On Furlough** | **Transition to Unemployment** |
| Intercept | −0.115 (0.810) | 3.952 *** (0.811) | 1.512 (0.929) | −0.951 (0.709) | 2.087 *** (0.546) |
| Age | −0.049 (0.037) | −0.178 *** (0.038) | −0.125 **(0.044) | −0.073 ** (0.031) | −0.105 *** (0.024) |
| Age 2 | 0.001 (0.000) | 0.002 *** (0.000) | 0.002 **(0.001) | 0.001 ** (0.000) | 0.001 *** (0.000) |
| Higher education | −0.499 *** (0.113) | −0.656 *** (0.137) | −0.814 *** (0.158) | −0.247 * (0.127) | −0.222 ** (0.112) |
| Finished education | 0.233 * (0.122) | 0.264 ** (0.124) | 0.368 ** (0.131) | −0.120 (0.114) | 0.308 *** (0.088) |
| Partner | −0.108 (0.101) | −0.265 ** (0.117) | 0.420 ** (0.135) | 0.041 (0.124) | −0.117 (0.103) |
| N° children 0 to 5 | −0.017 (0.115) | −0.193 (0.135) | 0.348 ** (0.135) | −0.135 (0.131) | −0.262 ** (0.109) |
| N° children 6 to 13 | 0.057 (0.100) | −0.181 (0.115) | 0.392 ** (0.122) | 0.038 (0.112) | 0.094 (0.095) |
| Main breadwinner | −0.054 (0.101) | −0.578 *** (0.123) | −0.517 *** (0.140) | −0.048 (0.124) | −0.449 *** (0.098) |
| Self-employed | 0.421 ** (0.199) | 0.163 (1.370) | 0.164 (1.410) | 0.018 (0.179) | 0.535 *** (0.124) |
| Part-time (<30 h) | −0.412 * (0.210) | −0.039 (0.212) | 0.251 (0.224) | 0.336 (0.224) | 0.093 (0.172) |
| Formal employment | 0.620 *** (0.173) | −0.484 ** (0.160) | −0.882 *** (0.177) | 0.315 ** (0.160) | −0.492 *** (0.122) |
| Contract | 0.171 (0.199) | −0.653 *** (0.181) | −0.727 *** (0.201) | −0.142 (0.172) | −0.526 *** (0.138) |
| Working hours | −0.022 *** (0.006) | −0.012 ** (0.006) | −0.012 ** (0.006) | −0.003 (0.005) | −0.013 **(0.004) |
| Years of seniority | 0.006 (0.007) | −0.054 *** (0.009) | −0.067 *** (0.011) | 0.000 (0.005) | −0.042 *** (0.005) |
| Medium-skilled occupation | 0.953 *** (0.126) | 0.419 ** (0.162) | 0.558 ** (0.196) | 0.858 *** (0.152) | 0.215 (0.131) |
| Low-skilled occupation | 0.852 *** (0.157) | 0.700 *** (0.180) | 0.473 ** (0.216) | 0.741 *** (0.192) | 0.272 * (0.155) |
| Primary sector | −2.163 *** (0.341) | 0.308 * (0.181) | 0.672 *** (0.189) | −0.964 *** (0.162) | 0.224 * (0.121) |
| Manufacture and services | −0.906 *** (0.184) | −0.183 (0.178) | −0.053 (0.193) | −0.170 (0.119) | 0.476 *** (0.106) |
| Commerce | −0.708 *** (0.130) | −0.291 ** (0.145) | −0.039 (0.155) | −0.245 * (0.143) | −0.191 (0.135) |
| Area: Rural | −0.154 (0.132) | −0.082 (0.133) | 0.204 (0.138) | −0.487 *** (0.132) | −0.230 *** (0.099) |
| Macro zone: North | 0.105 (0.135) | −0.365 ** (0.153) | 0.308 * (0.164) | −0.057 (0.141) | −0.083 (0.124) |
| Macro zone: South | 0.276 ** (0.102) | −0.160 (0.113) | 0.305 ** (0.134) | −0.108 (0.106) | 0.135 (0.091) |
| N | | 3981 | | | 4812 |
| R | | 0.159 | | | 0.106 |

Source: own elaboration based on ENE 2020 $p < 0.001$ ***, $p < 0.05$ **, $p < 0.01$ *.

As presented in the previous section, younger people are more likely to be in the unemployed cluster through 2020 for both men and women. This is in line with other studies that found that young workers were more susceptible to the economic effects of COVID-19 (Adams-Prassl et al. 2020; Falk et al. 2021), as they had accumulated fewer years of seniority for labor protection and access more precarious jobs than older generations. On the other hand, higher education is strongly correlated to remaining employed through 2020, especially for women. Good working conditions (no self-employment, labor formality, with contract) were also strongly associated with remaining employed throughout 2020 for both men and women. Occupational category became especially important for women, as women in low-skilled occupations had a greater likelihood to move into unemployment or care work than remaining employed.

Family characteristics played an important role in increasing the chance of transitioning into care work for women. Working women in a relationship and with small children were more likely to shift into care work than remaining employed throughout 2020. Conversely, men with young children were more likely to remain employed than belonging to the unemployed cluster. As we pointed out earlier, this would be strongly related to the male breadwinner culture present in Chile (Contreras and Plaza 2010; Murray 2015; Staab 2012), where it is the man who mostly assumes paid work, while women are dedicated to care work. This dynamic occurs particularly in periods of crises, reinforcing gender stereotypes, as has been reported in past economic crises (Karamessini and Rubery 2013; Sani 2018).

## 5. Conclusions

The present study was mainly exploratory, i.e., it sought to describe the working trajectories of workers during the early stages of the pandemic in Chile, instead of looking at the causal explanation of unemployment. Taking this into consideration, we can say that we met our hypotheses at the descriptive level. First, men remained in continuous employment in greater proportion than women. Second, male workers with family remained employed in a higher proportion than female workers with family. Third, at least half of the women who stopped working after the start of the pandemic did so for care-giving reasons, which would not be the case for men. In addition, women who moved into care work had the worst working conditions. As noted above, it is necessary to highlight the association between poorer working conditions, family responsibilities, and the transition to care work[15], which leaves unemployed women with young children economically dependent on their partners. Although in the present survey there is no data regarding the time spent on care work for men and women, other studies in Chile show that the gender gap in hours spent on household chores existed before the pandemic, and that that gap is widening[16]. Thus, the decline in women's growth in the labor market because of the pandemic could also be accompanied by a reinforcement of the male breadwinner model.

## 6. Discussion

This article explored the employment trajectories of women and men during the ongoing Coronavirus pandemic (COVID-19) in Chile, a neoliberal countries with a strong male breadwinner culture and high levels of income inequality. Considering the January to November period of the 2020 National Employment Survey, we studied how workers navigated through employment, furloughing, joblessness, and inactivity during one of the biggest economic crises that Chile has experienced in recent years.

The evidence showed that men in our sample lost their jobs to a lesser extent and returned to the labor market faster than women. This evidence is aligned with other studies on the impact of the Coronavirus on employment worldwide, where women have been the most affected (Adams-Prassl et al. 2020; Alon et al. 2020; Collins et al. 2020; Del Boca et al. 2020; Farré et al. 2020; Hipp and Bünning 2020; Reichelt et al. 2020). Men who kept their employment were mostly the household's main breadwinner, in a relationship, and with young children. On the other hand, working women who became inactive for family reasons were mostly in a couple, with children, and economically depended on their partners. The literature has already pointed out how economic crises can have adverse effects on progress towards gender equality, as was the case in Spain and Greece during the 2008 economic crisis (Alcañiz and Monteiro 2016; Hozic and True 2016; Sani 2018). European countries' austerity plans to cope with the recent economic crisis have severely cut social welfare and social programs that ensured gender equality and support for women (Hermann 2017; Karamessini and Rubery 2013). Chile is already getting a glimpse of how the crisis is affecting women the most, by (1) interrupting their working careers to a greater extent than men, and (2) reducing their pension funds, which were already lower than men's before the crisis. Chile must face the economic crisis with a gender perspective, considering the fragility of female employment and the strong male breadwinner culture that is present in a society that places the burden of care work mainly on women.

In our sample, around 30% of the workers worked informally (without social security or pension fund contributions), a percentage close to that estimated by INE's official statistics. Early research on the effect of the Coronavirus on employment pointed out that the crisis would not be experienced equally around the world, as many employees in developing economies work informally and cannot afford to work from home (Dingel and Neiman

---

[15] As noted by Maurizio and Monsalvo (2021), in Latin America informality is associated with a statistically significant earnings penalty, with lower-tier informal jobs being the lowest paid. This adds to the fact that, according to the authors, women experience a wage penalty in all work statuses, suggesting wage discrimination against them.

[16] Estudio Longitudinal Empleo COVID-19. *Centro UC Encuestas y Estudios Longitudinales*. Available online (only in Spanish): https://www.uc.cl/site/efs/files/11364/presentacion-estudio-empleo-covid19-13082020.pdf (accessed on 3 April 2021).

2020; Saltiel 2020). Indeed, evidence of this study shows that informal and self-employed workers, as well as workers without a contract, were the most affected by the pandemic. Only a small percentage of workers went on furlough, i.e., those protected by labor laws and contracts. This finding goes in the same direction as previous studies showing that self-employment is a manifestation of a historical framework of inequality of opportunities, and that self-employment in Latin America is a means of subsistence in response to unemployment and an economic stagnation trap (Rene Caceres and Caceres* 2017).

It should be noted that the transition from work to inactivity was not the same for men and women. For many men, inactivity meant unemployment and the constant search for work, while others reported not looking for work for various reasons (seasonal reasons, no desire to work, among others). On the other hand, for many women, inactivity meant taking over family and household duties. Previous research stated that the current pandemic has been especially detrimental in reinforcing the difference between paid and unpaid work among men and women (Del Boca et al. 2020; Hipp and Bünning 2020), not only for women who have withdrawn from the labor market to care for their children and/or relatives, but also for women who continued working from home. In our sample data, 22% of women in the continuously employed cluster still worked from home in the last wave (October and November 2020), compared to 8% of men in the same cluster. These women have to balance paid work with care work, in an environment of economic, labor, and health uncertainty.

The present study had certain limitations. First, it is an exploratory and descriptive study of the trajectories of women and men who were working before the pandemic and how their employment status changed throughout the year. The objective of this study was neither predictive nor causal, so we cannot generalize the findings to the total population. However, we believe that a comprehensive description of the trajectories of men and women analyzed can be a contribution to the study of how workers in the Global South are coping with this economic crisis, especially in a country with a rudimentary furloughing system and high labor informality. The increase in the level and persistence of unemployment after the pandemic supports the design and implementation of policies that minimize or avoid companies firing workers during a pandemic (Vladimir Rodríguez-Caballero and Vera-Valdés 2020), but unfortunately, in countries with high levels of informal work, these mechanisms are far from having a real impact on unemployment, as this research showed when analyzing furloughed workers. Future research will have to continue studying the causal effects of the economic crisis derived from the Coronavirus in the Global South.

This article contributes to the literature on economic crises, employment, and gender in two ways: first, using data that have not been widely used for academic purposes, it offers a first exploration of the effects of the economic crisis generated by the ongoing pandemic, and how these effects are more detrimental in a country with a small welfare state and high levels of inequality such as Chile. Following Blustein and Guarino (2020) recommendations, societies need to confront the institutions and norms that have created such profound levels of inequality, which has made the work-based crisis even more intense. In the case of Chile, not only are measures to increase economic wellbeing needed, but also policies to decrease inequalities.

Second, it establishes an interesting divide into how economic crises affect different types of employment, focusing mostly on its effects over informal employment within gig economies. This point is relevant since it offers a first approximation that can motivate other researchers into exploring the effects of the pandemic in other countries of the Global South, in a context where informal employment is often the norm and remains a consistent feature of work forces, which translates to workers having a subordinate role in their relations both with the state and through their insertion into formal production networks (Rogan et al. 2017). In more specific terms, exploring the effects of the current economic crisis on employment with a gendered lens, and putting women in the center of the study, provides a relevant insight not only for academic purposes but also from the perspectives of policymakers. Knowing the characteristics of the women who are being more severely

affected by the pandemic and the underlying reasons provides an insight that can orient future policies that aim to that group in particular.

**Author Contributions:** Conceptualization, V.R. and F.C.; methodology, V.R.; software, V.R.; validation, V.R. and F.C.; formal analysis, V.R. and F.C.; investigation, V.R. and F.C.; data curation, V.R.; writing—original draft preparation, V.R. and F.C.; writing—review and editing, V.R. and F.C.; visualization, V.R.; supervision, V.R. and F.C.; project administration, F.C. Both authors have read and agreed to the published version of the manuscript.

**Funding:** This research received no external funding.

**Informed Consent Statement:** Informed consent was obtained from the Instituto Nacional de Estadísticas, Chile, from all subjects involved in the study prior to data collection.

**Data Availability Statement:** Data available online: https://www.ine.cl/estadisticas/sociales/mercado-laboral/ocupacion-y-desocupacion (accessed on 16 April 2021).

**Acknowledgments:** We acknowledge support by the German Research Foundation (DFG) and the Open Access Publication Fund of Humboldt-Universität zu Berlin.

**Conflicts of Interest:** All authors declare no conflict of interest.

## Appendix A. Definition of the Number of Clusters

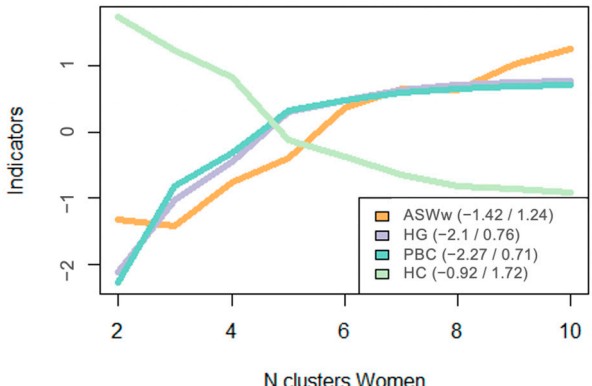

**Figure A1.** Definition of the number of clusters for women. Source: own elaboration based on ENE.

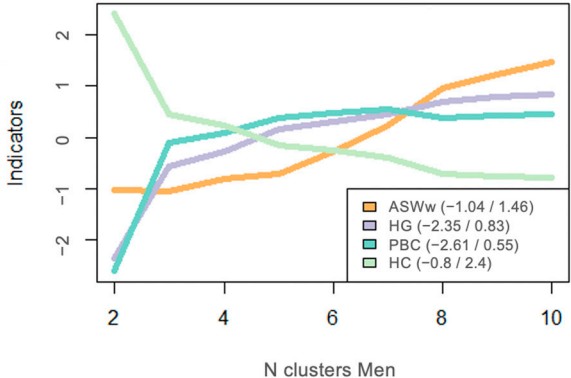

**Figure A2.** Definition of the number of clusters for men. Source: own elaboration based on ENE.

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
