# Peer review of "Between Social Protests and a Global Pandemic: Working Transitions under the Economic Effects of COVID-19"

_socsci, doi:10.3390/socsci10040145_

Round 1

Reviewer 1 Report

The article makes an exploratory analysis of the Chilean unemployment trajectories. The authors consider panel data from the Chilean National Employment Survey and show that the women and men employment trajectories differ. Men return to the labor market faster than women, which have repercussions for growth and the gender gap.

The results are interesting and can help in the design of policies to alleviate the economic effects of the pandemic. Nonetheless, there are some aspects of the analysis that I consider can be improved.

  • A little more clarification regarding the data would be useful. In particular, the phrase "we selected people" in line 197 is too vague and could be prompt to misunderstandings.
  • The exposition of the multinomial logistic regression can be improved. Particularly, it is not clear how the time-dimension was handled in the regressions.
  • Additional controls for the region could be considered.
  • I suggest adding some robustness exercises regarding the cost matrix. For instance, considering different costs for "on furlough" to "care work".
  • The chi-square test in Table 3 is not explained.
  • An additional robustness exercise could be conducted regarding the employment groups of men, Figure 6. The authors could consider merging the declining employment and transition to inactivity groups, as well as the short break and longer break groups to get more insight into the characteristics of these similar groups.
  • The paper would greatly benefit from some policy suggestions given the authors' results. The authors' could set their analysis in the context of exercises like Blustein et. al. (2020), Kouch et. al. (2020), Blustein and Guarino (2020)
  • The discussion on the long-term effects of the crisis on the labor market could be improved. It has been argued that pandemics have long-lasting consequences for unemployment (see Weinstock, L. R. (2020) Falk et al. (2021),  Rodriguez-Caballero and Vera-Valdes (2020)). The authors could comment on the long-term dynamics of unemployment if the dynamics studied were to continue unchanged

Minor comments:

  • The introduction refers to the crisis challenging "the responsiveness of neoliberal economies to social crisis". One could argue that it is not the responsiveness of "neoliberal economies".
  • The paper uses both underdeveloped and developing nomenclatures. I would suggest sticking with the most common developing term.
  • The term "mobile trimesters" in line 181.
  • The term "thick" in line 450.

References:

Blustein, D. L., & Guarino, P. A. (2020). Work and Unemployment in the Time of COVID-19: The Existential Experience of Loss and Fear. Journal of Humanistic Psychology, 60(5).

David L. Blustein, Ryan Duffy, Joaquim A. Ferreira, Valerie Cohen-Scali, Rachel Gali Cinamon, Blake A. Allan (2020) "Unemployment in the time of COVID-19: A research agenda", Journal of Vocational Behavior, 119, 103436.

Falk, G., Carter, J., Nicchitta, I., Nyhof, E., & Romero, P. (2021). "Unemployment Rates During the COVID-19 Pandemic: In Brief".Congressional Research Service, R46554

Kenneth A. Couch, Robert W. Fairlie, Huanan Xu (2020) "Early evidence of the impacts of COVID-19 on minority unemployment", Journal of Public Economics, 192, 104287.

Rodriguez-Caballero, C. V., & Vera-Valdes, J. E. (2020). Long-lasting economic effects of pandemics: Evidence on growth and unemployment.Econometrics,8

Weinstock, L. R. (2020). COVID-19: How Quickly Will Unemployment Recover?Congressional Research Service, IN11460

Author Response

Dear reviewers,

Before we provide a detailed report of the changes we have made to the manuscript, we would like to thank you for your feedback and the helpful suggestions on how to improve our paper. We addressed all concerns that were raised and followed the advice that was given.

In the following, we provide a point-by-point review and explain the changes we made. We quote your comments in italics; our replies follow right below in regular text.  

We believe that the revision process has substantially improved the manuscript and hope that you will deem the revised manuscript suitable for publication in the Special Issue of Social Sciences “Family, Work and Welfare: A Gender Lens on COVID-19”.

We look forward to your feedback.

Sincerely

The authors

Response to Reviewer #1

Comment 1: A little more clarification regarding the data would be useful. In particular, the phrase "we selected people" in line 197 is too vague and could be prompt to misunderstandings.

Reply: We were more clear in describing the case selection in order to avoid misunderstandings, as in the next paragraph: "To study the change in employment during 2020, we relied on a subsample of individuals of working age who declared to be working before the arrival of the first case of COVID-19 in Chile and were subsequently surveyed after the implementation of sanitary measures such as quarantines, curfews, store closures, and movement restrictions in public places. The main criteria for case selection were two (1) working people who were surveyed before the start of the pandemic in March (that is, surveyed during January of February 2020), and (2) who were successfully interviewed at least 3 additional times during 2020, totaling four waves. This resulted in an N of 8,864 cases (35,456 person-months). " We further included a table indicating the distribution of cases.

Comment 2: The exposition of the multinomial logistic regression can be improved. Particularly, it is not clear how the time-dimension was handled in the regressions.

Reply: Multilevel models do not consider the time dimension because they measure the probability of membership to a working trajectory. We make the steps of our plan analysis more clear by dividing the section into two parts. First, we explain that we performed the sequences analysis (which includes the time dimension). And then, we explain the multilevel analysis that predicts the probability of falling in each of the clusters previously defined. The last analysis only includes demographic variables.

Comment 3: Additional controls for the region could be considered.

Reply: We included additional controls of region (north, central, south of the country)

Comment 4: I suggest adding some robustness exercises regarding the cost matrix. For instance, considering different costs for "on furlough" to "care work".

Reply: We performed the robustness exercise and changed the cost matrix, only leaving a higher cost for 'on furlough' and the transition to other categories, as explained in the next paragraph "We constructed a custom cost matrix, where we included a higher cost to transit from On furlough to the other categories and vice-versa, as "On furlough" is a special category of those who are employed, and it is more costly for employers to fire someone protected by the furlough. We tested different cost matrixes as a robustness exercise (not present in this article but available upon request) and found that the clusters with the best measures fit and explanatory potential for the questions proposed by the present work were derived from the matrix used."

Comment 5: The chi-square test in Table 3 is not explained.

Reply: We added the corresponding explanation: "The chi-squared test in the third column measures the independence between the exposed variables and gender. It is reported if there is an association between gender and the corresponding variable, at the different confidence levels specified. "

Comment 6: An additional robustness exercise could be conducted regarding the employment groups of men, Figure 6. The authors could consider merging the declining employment and transition to inactivity groups, as well as the short break and longer break groups to get more insight into the characteristics of these similar groups.

Reply: We redid the cluster analysis focusing more on the workers who transitioned to the same categories rather than people who shared the same timing of transition (as we did in the first cluster analysis). Thus, we were left with 4 clusters for women and 3 for men and were able to get a better insight into the characteristics of these groups.

Comment 7: The paper would greatly benefit from some policy suggestions given the authors' results. The authors' could set their analysis in the context of exercises like Blustein et. al. (2020), Kouch et. al. (2020), Blustein and Guarino (2020)

Reply: There was a lack of policy suggestions indeed. Reference to Blustein & Guarino (2020) was added in line 518; Blustein et al. (2020) from lines 90-94 regarding the usefulness of longitudinal data in order to identify unemployed groups or clusters; Couch et al. (2020) reference added in 131-133 to support the trend that unemployment during the pandemic affected those with a lower educational level on a bigger extent. 

Comment 8: The discussion on the long-term effects of the crisis on the labor market could be improved. It has been argued that pandemics have long-lasting consequences for unemployment (see Weinstock, L. R. (2020), Falk et al. (2021), Rodriguez-Caballero and Vera-Valdes (2020)). The authors could comment on the long-term dynamics of unemployment if the dynamics studied were to continue unchanged

Reply: Reference to Weinstock (2020) was added in line 219 related to the uncertainty of the effects of the pandemic. Reference to Falk et al. (2020) was added in lines 133-136 to reinforce the trend that women are affected, to a bigger extent, by unemployment during this pandemic. Reference to Rodriguez-Caballero & Vera-Valdes (2020) was added in lines 507-513.

Comment 9: The introduction refers to the crisis challenging "the responsiveness of neoliberal economies to social crisis". One could argue that it is not the responsiveness of "neoliberal economies".

Reply: Given the following sentence, when countries as the US, Brazil, and the UK are mentioned, we stand by the argument that the poor responsiveness to the crisis that emerged due to the pandemic is indeed related to a neoliberal understanding of social and economical issues.

Comment 10: The paper uses both underdeveloped and developing nomenclatures. I would suggest sticking with the most common developing term.

Reply: Change made to "developing".

Comment 11: The term "mobile trimesters" in line 181.

Reply: Changed to "mobile quarters".

Comment 12: The term "thick" in line 450.

Reply: Changed to "a comprehensive description"

Additional changes

As we noted earlier, we remade the cluster analysis, prioritizing the gathering of workers in groups that shared similar transitions rather than workers who shared the same timing of transitions (as we only had four waves and the groups of Short break, Long break, etc, contained very few cases for a robust analysis). We also remade the tables and figures to simplify the gender comparison.  

Reviewer 2 Report

See attached document (pdf)

Author Response

Dear reviewers,

Before we provide a detailed report of the changes we have made to the manuscript, we would like to thank you for your feedback and the helpful suggestions on how to improve our paper. We addressed all concerns that were raised and followed the advice that was given.

In the following, we provide a point-by-point review and explain the changes we made. We quote your comments in italics; our replies follow right below in regular text.  

We believe that the revision process has substantially improved the manuscript and hope that you will deem the revised manuscript suitable for publication in the Special Issue of Social Sciences “Family, Work and Welfare: A Gender Lens on COVID-19”.

We look forward to your feedback.

Sincerely

The authors

Response to Reviewer #2

Comment 1: The first suggestion is related to the title, considering two aspects.

On the one hand, throughout the article, there is no analysis on the social protests. Neither its causes, nor its consequences. It is taken for granted in the title that the labour market context (specially, the employment) prior to the pandemic in Chile is due to the impact of the social outbrake. However, most Latin American countries have high unemployment and informality levels and weak socioeconomic conditions, without having suffered social protests. In any case, this cannot be demonstrated with the literature used in the paper (cannot be found in it quotes or bibliography on this issue), and besides, data for the model considers only 2020, not the previous period. Nevertheless, I think it should not be necessary to include any reference on social protests, since the research is focus on the pandemic impact on labour trajectories during 2020.

On the other hand, the object of the analysis is employment, on furlough, and inactivity, but little is said about unemployment. The model tackles employment and transitions, both for male and female workers, but there is no explanation on women’s unemployment.

I do recommend the authors improving the title (social protests and unemployment concepts might not be included) and adding the key words of the research for a better understanding of the paper, for instance: labour trajectories, comparing women and men employment and transitions, and global pandemic COVID-19.

Reply: The title was meant to locate the readers into the Chilean context, not to make a causal claim that the protests are the cause of the economic landscape at the beginning of 2020. Even when it is true that the data of the model considers only 2020, protests continued to happen during that year and in 2021. We believe that making an analysis of labor trajectories, ignoring the context of social unrest that has been ongoing since October 2019, would be a mistake. However, we took the suggestion made and changed the name of the paper to emphasize the exploratory nature of the paper, as well as its focus on labor transitions rather than unemployment. The final title is: “Between social protests and a global pandemic: working transitions under the economic effects of COVID-19”.

Comment 2: The second suggestion deals with the potential academic contribution to the literature. It is of real interest the empirical analysis of the paper and its contribution to the understanding of labour market performance in Chile, during the pandemic, and from a gender view. However, the results of the article have the opportunity to contribute to the socioeconomic literature within other relevant approaches of labour socioeconomics.

Human capital theories, concerning the role of education and training in the labour market. As far as the results of the model sustain the paramount role of these variables in employment.

Job quality or decent work concepts (based in UN or ILO sources), with reference to formal contracts, full-time, working hours, occupations,.. In this sense, the segmentation and dual labour markets works, within the institutionalist approach, could be useful for further debate.

Thus, apart from the totally appropriate gender literature references (some comments will be added in 2.1), one (or both) of the two areas of research mentioned may be presented in section 2, and discussed after the model outcomes in section 5.

Requested changes could be done to target this aim on the theoretical relevance of the article.

    Reply: In subsequent, more specific comments, we present the additional changes that address the topic of human capital theories and job quality. 

Comment 3: The last suggestion regards the lack of research questions (or hypothesis). Once the authors may have considered including the approach(es) above, it is recommended posing some research questions on the stage, to cope with the second suggestion and introduce some debate elements in the article.

Reply: We included our research questions and hypotheses in more detail during the theoretical exposition of the article. We also included a Conclusions section which links our results with the proposed hypotheses.

Comment 4: 13: duplicity of the concepts: economic recession, economic crisis.

Reply: Economic crisis was changed for "unemployment during crisis"

Comment 5: 49-51: The quote refers to 2010. Could be appropriate to include a mention or more recent data of the variable.

Reply: Two more recent reports were added: one from the National Institute of Statistics (2015), and the most recent one from CEPAL/ECLAC (2021).

Comment 6: 54-55: “..and the pandemic is further deepening the distance between the precarious and well-off workers, and between working women and men.” The latter needs further study after analysing (separately) women and men trajectories in the article. For example: 298-306 and 355-361: Trajectories for men and women could be compared with more details, in order to obtain gender gap conclusions. For instance, “Continuously employed” characteristics seem to be similar in men and women.

Reply: We cite a recent study showing that during 2020 in Chile the female labor force declined by 41% and the pandemic is expected to be a reversal of 10 years of gains for women (see footnote 3). On the other hand, for a better comparative analysis of the trajectories of men and women, we combined the separated tables and figures into a single one, where the comparison is straightforward (see Table 5, and 6, as well as Figure 4). Likewise, during the description of the trajectories, we highlighted the difference between men and women (see Page 15).

Comment 7: Section 2.1. A framework on the opposite effects showed during the different phases of the economic cycle (expansion or recession) regarding discouraged workers and additional workers, comparing gender tendencies, can be useful. Some examples in Asia are introduced in the text, that could be complemented with Western economies cases. For instance, in some UE15 countries, during the Great Recession, female workers accessed the labour market, increasing the activity rates for women, whereas men were more affected by the discouragement effect, decreasing their activity rates (this can be used to complete the debate ideas in lines 120-121). On the contrary, during the former economic expansion, before 2008, could be found the opposite behaviour. The explanation is related to the economic activities involved in that crisis (construction sector and industries, masculinised economic branches), that resulted in a greater employment negative impact for men. See more detailed in: Employment Quality And Gender Equality. An Analysis For The European Union. Regional and Sectoral Economic Studies, 2020. Volume 20, Issue 2, 2020.

Reply: Reference to Pérez Ortiz et al. (2020) was added in lines 150-152 to reinforce the fact that women tend to occupy more precarious jobs. Additionally, lines 328-330 were added to make reference to the type of elasticity of female employment and how it is affected by, for example, caregiving labor. 

Comment 8: 109-114: In the current context, in Europe, female-dominated sectors have performed better, such as health, education, care sector (mainly essential activities). This reflects a difference comparing to the last crisis. Did it happen in Chile too? If so, include a reference to complete the ideas.

Reply: Latest official data available (from 2015, provided by the National Institute of Statistics) states that half of the employed women are employed in Non-qualified jobs or the Service Sector. Within the Professional sector, there are no trends to make such statement that areas as education or health are female-dominated sectors. Caring labor is usually carried by women but in an informal manner. This information was added into that paragraph to clarify.

Comment 9: 122-126: It is of interest for the paper’s discussion to look further into the implications of remote work and women caregivers role during the lockdowns, reflecting that although more educated female workers with better job quality conditions (blue collar) kept their jobs (as men did), they were more involved in unpaid work at home. A fact clearly stated in Europe by Eurofound | (europa.eu) in its Living, working and COVID-19 e-survey.

Reply: Unfortunately, we do not have information about the amount of care work at home performed by women who remained employed. We can only make a reflection on this regard at the end of the discussion.

Comment 11: Finally, another aspect to be taken into account is job quality gender gaps, under the segmentation theoretical approach. There are enough data information to answer to research questions on this issue too.

Reply: While there are no major gender gaps in job quality between men and women in our sample (Table 2, Descriptive statistics). Job quality becomes more important for women when it comes to becoming unemployed during the pandemic. One of our hypotheses is related to this, as we hypothesized that in Chile it will be women in low-skilled occupations the most affected by unemployment during the pandemic.

Comment 12: 147-148: “During the 21st century, Chile has not experienced major economic crises, maintaining a steady economic growth and reducing its poverty rate.” This is in contradiction with the idea that social protests changed the socioeconomic context prior the pandemic. It is needed to clarify the authors’ position, in case the social protests subject is finally maintained in the text.

Reply: We did not say that protests changed the socioeconomic context before the pandemic, but that inequality ignited the protests of 2019. Lines 167-172 added more information about inequality levels in Chile, and footnote 4 further explains the origins of the protests.

Comment 13: 215: It is required an explanation (in a footnote could be possible) of the cost concept (Direct or indirect? Opportunity costs for workers? Paid by employers?)

Reply: Cost refers to the theoretical cost of moving from one position to the other. There is not only one way to establish the cost in the sequence analysis, being left to the definition of the researcher based on a theoretical criterion. For this, we tried different costs matrix, remaining with the final matrix proposed, as detailed in the following paragraph: “We constructed a custom cost matrix, where we included a higher cost to transit from On furlough to the other categories and vice-versa, as "On furlough" is a special category of those who are employed, and it is more costly for employers to fire someone protected by the furlough. We tested different cost matrixes as a robustness exercise (not present in this article but available upon request) and found that the clusters with the best measures fit and explanatory potential for the questions proposed by the present work were derived from the matrix used.”

Comment 14: 232: Figure 3, instead of 2

Reply: We re-did the figure and tables and now there is a different numbering.

Comment 15: 233-236: Further information of the clusters is needed. Their composition and the information embedded: is it the same for male and female workers?

Reply: We redid the cluster analysis focusing more on the workers who transitioned to the same categories rather than people who shared the same timing of transition (as we did in the first cluster analysis). Thus, we were left with 4 clusters for women and 3 for men and were able to get a better insight into the characteristics of these groups. That is, three clusters for men and women: 1. Continuously employed, 2. On furlough, 3. Transition to unemployment, with an additional group for women 4. Transition to care work.

Comment 16: 268-269: “It is important to note that women mainly work in occupations with higher skills (i.e. managers, professionals, and technicians) in higher proportions than men do”. This result is surprising in the context of vertical segregation challenges. The glass ceiling approach supports, to a certain point, the opposite. Thus, this fact needs to be explained with detailed (it is recommended consulting some references).

Reply: We understand that the information regarding women accessing occupations with higher skills could have been misleading, in the sense that they are located into the upper part of the earnings distribution, but this is not the case. In lines 288-294 we make this clarification, pointing to the fact that in Chile the wage gap is still substantial, and that women don't access to top management positions in the same proportion as men. The main reason why women access to occupations with higher skills is that they enter higher education in a bigger proportion. 

Comment 17: 270: horizontal segregation.

Reply: added "an horizontal segregation that goes in the same..."

Comment 18: 286-294: It is suggested a deeper description of the clusters, specially “Declining employment”.

Reply: As stated in the reply to comment 15, we redid the cluster analysis and deepened the description.

Comment 19: 452-453: it is mentioned “a rudimentary furloughing system”. The observation needs to be clarified with a brief description of this furloughing system (preferably before, in a descriptive section, or a footnote).

Reply: Explanation added on foot note 12, when the concept is first introduced.

Comment 20: 298-306 and 355-361: Trajectories for men and women could be compared with more details, in order to obtain gender gap conclusions. For instance, “Continuously employed” characteristics seem to be similar in men and women. This discussion could enrich the debate, under the theoretical approaches mentioned in the General comments.

Reply: With the new clusters of employment trajectories, we described the path themselves (e.g. Continuously employed) and highlighted the gender differences between the cluster (see Page 15). This way, even if men and women are clustered into similar groups, we analyzed the difference within these groups with a gender approach. 

Comment 21: The empirical results could be contrasted with job quality/decent work and segmentation of labour markets approaches, as well as human capital theories, as indicated in the General comments section.

Reply: Footnote number 15 was added to reinforce the fact that in Latin America, the working conditions of women are significantly lower than men. Additionally, lines 144-146 address the matter as well.

Comment 22: Appendix. May consider moving some tables or figures to the appendix.

Bibliography section must be reviewed. There are some incomplete references. i.e.: 487, 514, 519, 546, 576, 606,... 

Reply: Bibliography section completed; missing information added and other formatting problems solved (specifically regarding capitalization of names and countries)

Additional changes

As we noted earlier, we remade the cluster analysis, prioritizing the gathering of workers in groups that shared similar transitions rather than workers who shared the same timing of transitions (as we only had four waves and the groups of Short break, Long break, etc, contained very few cases for a robust analysis). We also remade the tables and figures to simplify the gender comparison.  

Round 2

Reviewer 1 Report

The paper has improved since its initial submission. I thank the authors for addressing the previous feedback.

I maintain that the crisis has challenged the responsiveness of all countries, not just the neoliberal ones. The authors should present arguments or citations to other studies that find neoliberal countries to be hit differently than the rest, or explicitly state that the study only deals with a neoliberal country and cannot be extrapolated to non-neoliberal ones.

Minor comment: Typo in line 451.

Author Response

Dear reviewers,

Thanks for this second round of revisions. In the following, we provide a point-by-point review and explain the changes we made. We quote the reviewers’s commentaries in italics; our replies follow right below in regular text. 

We believe that the revision process has substantially improved the manuscript and hope that you will deem the revised manuscript suitable for publication in the Special Issue of Social Sciences “Family, Work and Welfare: A Gender Lens on COVID-19”.

Sincerely

The authors

Response to Reviewer #1, Round 2

Comment 1: I maintain that the crisis has challenged the responsiveness of all countries, not just the neoliberal ones. The authors should present arguments or citations to other studies that find neoliberal countries to be hit differently than the rest, or explicitly state that the study only deals with a neoliberal country and cannot be extrapolated to non-neoliberal ones.

Reply: Changes made from lines 24-29, removing “and the responsiveness of neoliberal economies to social crises”. This idea was reformulated into a broader sense. Now, the paragraph states “The present crisis has particularly hit countries with precarious protection systems, such as the U.S., Brazil, and the U.K., deepening existing inequalities of race, class and gender (Blundell2020, Nunes.2020, Ortega.2020). State aid has been one of the main policies to help unemployed workers buffer the economic consequences of the pandemic, even in typically non-interventionist states, such as the U.S.”

Comment 2: Minor comment: Typo in line 451.

Reply: The structure of the sentence was changed. Now the line is “ Men who kept their employment were mostly the household’s main breadwinner, in a relationship, and with young children.”